# Polynomial-Time Fraïssé Limits via Fixed Point Theorems:
# From Model Theory to Trustworthy AI

**Andrey Nechesov**[1,3,4*] **& Vadim Puzarenko**[1,2,3]

[1]*Artificial Intelligence Center, Novosibirsk State University, 630090 Novosibirsk, Russia*

[2]*Sobolev Institute of Mathematics, SB RAS, 630090 Novosibirsk, Russia*

[3]*International Artificial Intelligence Committee, (IAIC), Dubai, UAE*

[4]*Russian Engineering Academy (IAE), Moscow, Russia*

[*]*Corresponding author: nechesoff@gmail.com*

## Abstract

We establish sufficient conditions under which the Fraïssé limit of a countable chain of finite p-computable algebraic structures is itself a p-computable structure. Our approach synthesizes two pillars: the polynomial analogue of Gandy's fixed point theorem (PAG-theorem), which guarantees that inductively defined predicate extensions remain polynomially decidable, and its functional variant (FPAG-theorem), which ensures that recursively defined functions on hereditary-finite list superstructures preserve polynomial complexity bounds. The central result—the *P-Fraïssé Theorem*—provides a unified criterion involving a $\Delta_0^p$-operator for generators, a $\Delta_0^p$-operator for predicates, and a functional boundary condition for operations. We demonstrate that classical Fraïssé limits—the random graph (Rado graph), the countable dense linear order without endpoints ($\eta$-order), the countable atomless Boolean algebra, and the countable Ershov algebra—all admit p-computable presentations under natural encodings. These results bridge classical model-theoretic constructions with the computational complexity requirements of trustworthy artificial intelligence, digital twin infrastructures for smart cities, and formally verified decision-support systems, where polynomial computational complexity serves as a fundamental reliability criterion.

**Keywords:** Fraïssé limits; polynomial computability; Gandy's fixed point theorem; model theory; trustworthy AI; digital twins; smart cities

**MSC 2020:** 03C13 · 03D15 · 68Q17 · 03C35

## 1 Introduction

Fraïssé's theorem (1; 2) occupies a central position in model theory, providing a canonical method for constructing countable homogeneous structures as limits of directed systems of finite structures. The *random graph* (Rado graph) (3; 4), the *countable dense linear order without endpoints* ($\mathbb{Q}$-order), and the *countable atomless Boolean algebra* all arise as Fraïssé limits of appropriate classes, establishing deep connections between combinatorics, algebra, and logic.

In parallel, the theory of polynomial-time (p-computable) structures (11; 12; 10) has matured into a rich area at the intersection of computable model theory and computational complexity. A fundamental question is: *which classical constructions from model theory can be realized within polynomial time?* This question acquires practical urgency in the context of modern artificial intelligence, where decision-support systems, digital twins of urban infrastructure (17), and smart contract verification (18) demand not merely computability, but computability within strict resource bounds. Polynomial computational complexity is not merely a theoretical desideratum but a **fundamental reliability criterion** for any AI system that must respond to real-time queries, manage critical infrastructure, or interact with physical environments where unbounded computation times are unacceptable (16).

The polynomial analogue of Gandy's fixed point theorem (PAG-theorem) (7) and its functional variant (FPAG-theorem) (8) provide the essential technical machinery for establishing polynomial-time bounds on inductively defined objects. The classical Gandy theorem (5; 6) states that the smallest fixed point of a positive $\Sigma$-operator is a $\Sigma$-set—which in general is merely semi-decidable, not polynomially decidable. The PAG-theorem overcomes this limitation by introducing $\Delta_0^p$-operators whose smallest fixed points are provably $\Delta_0^p$-sets. The FPAG-theorem extends this paradigm from predicates to functions, identifying sufficient conditions for recursively defined functions to remain within polynomial complexity bounds.

In this paper, we bring these threads together to prove our central result:

> **P-Fraïssé Theorem.** *Under natural effectiveness and boundedness conditions, the Fraïssé limit of a countable chain of p-computable finite structures is itself a p-computable structure.*

The significance of this result is threefold. First, **for mathematicians**, it provides a general polynomial-time realization theorem for a fundamental model-theoretic construction, complementing the classical theory of computable Fraïssé limits (2; 15) with explicit complexity bounds. Second, **for computer scientists**, it establishes that a wide class of inductively constructed infinite structures—precisely those arising in formal verification, type theory, and knowledge representation—admit efficient algorithms. Third, **for AI practitioners**, it ensures that hierarchical data structures modeling real-world systems (cities, transportation networks, organizational hierarchies) can be processed in guaranteed polynomial time, a prerequisite for trustworthy AI.

The paper is organized as follows. Section 2 recalls the necessary background on p-computability, the PAG-theorem, and the FPAG-theorem. Section 3 develops the theory of p-computable Fraïssé limits, culminating in the P-Fraïssé Theorem and its variants. Section 4 presents four concrete corollaries establishing p-computability for classical Fraïssé limits. Section 5 discusses applications to artificial intelligence, digital twins, and decision-support systems. Section 6 discusses broader mathematical significance and open problems.

## 2 PRELIMINARIES

### 2.1 POLYNOMIAL COMPUTABILITY

Let $\Sigma$ be a finite alphabet and $\Sigma^*$ the set of finite words over $\Sigma$. Throughout, we use the hereditary-finite list encoding: elements are either atoms from a base set $M \subseteq \Sigma_0^*$ (where $\Sigma_0 \subset \Sigma$) or lists $\langle a_1, \dots, a_n \rangle$ of previously constructed elements.

**Definition 2.1** (p-Computability (14; 11)). A function $f : A \to B$ (where $A, B \subseteq \Sigma^*$) is *p-computable* if there exist constants $C, p \in \mathbb{N}$ and a deterministic Turing machine $T$ such that for all $a \in A$, the value $f(a)$ is computed in at most $C \cdot |a|^p$ steps, where $|a|$ denotes the length of the word $a$.

A set $A \subseteq \Sigma^*$ is *p-computable* (or a $\Delta_0^p$-set) if its characteristic function $\chi_A : \Sigma^* \to \{0, 1\}$ is p-computable.

A model $\mathfrak{M}$ of finite signature $\sigma$ is a *p-computable model* ($\Delta_0^p$-model) if its base set and all predicates are $\Delta_0^p$-sets, and all functions are $\Delta_0^p$-functions.

**Definition 2.2** (Rank and Word Splitting (7)). The *word splitting function* $R : \Sigma^* \to (\Sigma \cup \{\#\})^*$ decomposes a list $\langle w_1, \dots, w_n \rangle$ into $w_1 \# \cdots \# w_n$, and is undefined on non-list elements. The *rank* $\mathrm{r}(w)$ of an element $w \in \Sigma^*$ is defined inductively:

$$\mathrm{r}(w) = \begin{cases} 0, & \text{if } R(w) \text{ is undefined,} \\ \sup\{\mathrm{r}(w_1), \dots, \mathrm{r}(w_k)\} + 1, & \text{if } R(w) = w_1 \# \cdots \# w_k. \end{cases}$$

The word splitting function $R$ is itself a $\Delta_0^p$-function with $t(R(w)) \leq |w|$. Crucially, $\mathrm{r}(w) < |w|$ for all $w$.

### 2.2 HEREDITARY-FINITE LIST SUPERSTRUCTURE

Given a p-computable model $\mathfrak{M}$ of signature $\sigma_0$, the *hereditary-finite list superstructure* $\mathrm{HW}(\mathfrak{M})$ (7; 9) is the structure whose base set $\mathrm{HW}(M)$ consists of all hereditary-finite lists over $M$, equipped with operations:

- $\mathrm{head}(x)$, $\mathrm{tail}(x)$—taking/removing the last element;
- $\mathrm{cons}(x, y)$, $\mathrm{conc}(x, y)$—appending an element and concatenating lists;
- $x \in y$, $x \subseteq y$—membership and initial-segment relations;
- $\mathrm{nil}$—the empty list constant.

### 2.3 THE PAG-THEOREM

The polynomial analogue of Gandy's fixed point theorem (PAG-theorem) (7; 9) is concerned with the inductive extension of predicates.

**Definition 2.3** (Generating Families and $\Delta_0^p$-Operators (7)). Let $HW(\mathfrak{M})$ be a model of finite signature $\sigma$ with unary predicates $P_1, \dots, P_n$. A *generating family* $\mathcal{F}_{P_i^+}$ is a finite or countable collection of quantifier-free, predicate-separable formulas of $\sigma$ with potive entry of predicates.

The operator $\Gamma^{HW(\mathfrak{M})}_{\mathcal{F}_{P_1^+}, \dots, \mathcal{F}_{P_n^+}}$ is called a $\Delta_0^p$-*operator* if:

1. $HW(\mathfrak{M})$ is a $\Delta_0^p$-model;

2. All formulas in $\mathcal{F}_{P_i^+}$ are positive, quantifier-free, and predicate-separable;
3. The generating formula for any element is unique;
4. The functions $\gamma_i$ constructing generating formulas are $\Delta_0^p$-functions, and truth-checking of formulas from generating families is p-computable.

**Theorem 2.4** (PAG-Theorem (7))**.** *The smallest fixed point $\Gamma^\omega$ of a $\Delta_0^p$-operator $\Gamma_{\mathcal{F}_{P_1^+},\ldots,\mathcal{F}_{P_n^+}}^{HW(\mathfrak{M})}$ is a $\Delta_0^p$-set.*

### 2.4 The FPAG-Theorem

The functional variant (FPAG-theorem) (8) addresses inductively defined *functions* rather than predicates.

**Definition 2.5** (FGNF-System (8))**.** A *p-computable FGNF-system* consists of:

- A p-computable hereditary-finite list superstructure $\langle HW(\mathfrak{M}), \sigma \rangle$;
- Initial p-computable functions $f_1, \ldots, f_n$ with functional boundary condition $|f(a)| \leq |a|$;
- Generative families of terms $\mathcal{T}_{f_i} = \{t_j(x_1, \ldots, x_{k_j}) \mid j \in \mathbb{N}\}$ with complexity boundary conditions;
- p-computable functions $\gamma_i : HW(M) \to \mathcal{T}_{f_i} \cup \{false\}$;

**Theorem 2.6** (FPAG-Theorem (8))**.** *Let a p-computable FGNF-system with initial p-computable functions $f_1, \ldots, f_n$ be given. Then the smallest fixed point $\Gamma^* = (f_1^*, \ldots, f_n^*)$ of the operator $\Gamma_{f_1,\ldots,f_n}^{HW(\mathfrak{M})}$ is p-computable.*

## 3 P-Computable Fraïssé Limits

### 3.1 Classical Context

Recall that in classical model theory, a *Fraïssé class* $\mathcal{K}$ is an age (class of all finite structures embeddable into a given structure) satisfying the *hereditary property* (HP), *joint embedding property* (JEP), and *amalgamation property* (AP). The *Fraïssé limit* $\mathfrak{B} = \mathrm{Flim}(\mathcal{K})$ is the unique (up to isomorphism) countable ultrahomogeneous structure whose age equals $\mathcal{K}$ (1; 2).

An alternative, constructive perspective—which is our starting point—is to realize the Fraïssé limit as the direct limit of a chain:

$$\mathfrak{M}_0 \hookrightarrow \mathfrak{M}_1 \hookrightarrow \mathfrak{M}_2 \hookrightarrow \cdots$$

where each $\mathfrak{M}_i$ is a finite structure and the embeddings satisfy the extension property. Our goal is to identify conditions under which this chain construction yields a p-computable limit.

### 3.2 Setup and Definitions

Let $\sigma = \{P_1, \ldots, P_k, f_1, \ldots, f_n\}$ be a finite signature with predicates and function symbols.

**Definition 3.1** (p-Computable Chain)**.** A *p-computable chain* is a countable family $\{\mathfrak{M}_i\}_{i \in \mathbb{N}}$ of finite $\sigma$-structures together with embeddings $e_i : \mathfrak{M}_i \hookrightarrow \mathfrak{M}_{i+1}$ such that:

1. Each $\mathfrak{M}_i$ is a p-computable finite structure (with all operations and predicates p-computable on the finite base set $M_i$).
2. Each embedding $e_i$ is an *inner embedding*: the image $e_i(M_i)$ forms a substructure of $\mathfrak{M}_{i+1}$.

**Definition 3.2** (Generators, Closure and Canonical Elements)**.** For a structure $\mathfrak{M}_j$ with base set $M_j$, we define:

- The *set of generators* $G(\mathfrak{M}_j) = \{g \in M_j \mid g$ is not in the range of any function $f_i$ applied to elements of $M_{j-1}\}$ (with $M_{-1} = \emptyset$), that is, the "new" elements introduced at stage $j$.
- The *closure* $\mathrm{Cl}(G(\mathfrak{M}_j))$ in signature $\sigma$ is the set of all inductivelly definable term-encoded elements $\underline{t_m} = \langle \underline{f_i}, \underline{t_1}, \ldots, \underline{t_{n_m}} \rangle$ where $f_i(t_1, \ldots, t_{n_m})$ is a term of $\sigma$ and $\underline{f_i}$ denotes the encoding of functional symbol $\underline{f_i}$.

- The *cumulative set of generators* at step $i$ is $S_i = \bigcup_{j \leq i} G(\mathfrak{M}_j)$, and $S = \bigcup_{i \in \mathbb{N}} S_i$.

- *Canonical Elements:* such elements that are either elements from $S$ or are special encoding that are canonical for a set of codes for each element. By default, we assume that the value of any term $t(\overline{a})$ is canonical.

- The *set of canonical elements* is denoted at step $i$ is $Canonical_i = \bigcup_{j \leq i} Can(\mathfrak{M}_j)$, and $Can = \bigcup_{i \in \mathbb{N}} Can_i$.

**Definition 3.3** (Functional Boundary Condition)**.** The chain $\{\mathfrak{M}_i\}_{i \in \mathbb{N}}$ satisfies the *functional boundary condition* if for each structure $\mathfrak{M}_i$ and each function $f_j \in \sigma$ is boundary:

$$|f_j(\bar{a})| \leq |\mathrm{list}(\bar{a})|$$

for all $\bar{a} \in M_i^{m_j}$, where $\mathrm{list}(a_1, \ldots, a_n) = \langle a_1, \ldots, a_n \rangle$.

This condition is essential: it ensures that the output of function applications does not grow faster than the input encoding, which is precisely what the FPAG-theorem requires to maintain polynomial bounds under recursive evaluation.

**Definition 3.4** ($\Delta_0^p$-Operator for Generators). A $\Delta_0^p$-*operator for the generator family* is a $\Delta_0^p$-operator $\Gamma_S$ (in the sense of Definition 2.3) such that $\Gamma_S(S_i) = S_{i+1}$ for all $i \in \mathbb{N}$, implemented via a generative family of $\Delta_0$-formulas that specifies how new generators are produced from existing ones at each stage.

**Definition 3.5** ($\Delta_0^p$-Operator for Predicates). A $\Delta_0^p$-*operator for predicates* is a $\Delta_0^p$-operator $\Gamma_{P_1,\ldots,P_k}$ (in the sense of Definition 2.3) that inductively extends the truth sets of predicates $P_1, \ldots, P_k$ from $\mathfrak{M}_i$ to $\mathfrak{M}_{i+1}$ as new elements are generated.

### 3.3 THE P-FRAÏSSÉ THEOREM

We now demonstrate that four fundamental Fraïssé limits admit p-computable presentations. In each case, we explicitly construct the chain, the encoding, the generative family, and the predicate operator, and verify all conditions of Theorem 3.6 (or Remark 3.7 for relational signatures).

**Theorem 3.6** (P-Fraïssé Theorem, Full Signature). *Let* $\{\mathfrak{M}_i\}_{i \in \mathbb{N}}$ *be a p-computable chain of finite algebraic structures of signature* $\sigma = \{P_1, \ldots, P_k, f_1, \ldots, f_n, Can^{(1)}\}$ *with Fraïssé limit* $\mathfrak{B} = \varinjlim \mathfrak{M}_i$. *Suppose:*

1. *(**Inner embedding**) Each* $e_i : \mathfrak{M}_i \hookrightarrow \mathfrak{M}_{i+1}$ *is an inner embedding.*

2. *(**Functional boundary**) The chain satisfies the functional boundary condition (Definition 3.3).*

3. *(**Generator operator**) There exists a* $\Delta_0^p$*-operator* $\Gamma_S$ *for the family of generators sets* $\{S_i\}$.

4. *(**Predicate operator**) There exists a* $\Delta_0^p$*-operator* $\Gamma_{P_1,\ldots,P_k}$ *for the predicates.*

5. *(**Canonical operator**) There exists a* $\Delta_0^p$*-operator* $\Gamma_{Can}$ *for the canonical predicates* $\{Can_i\}$.

*Then the Fraïssé limit* $\mathfrak{B}$ *is a p-computable structure.*

*Proof.* We must establish four properties: (a) the base set $B$ of $\mathfrak{B}$ is a p-comutable; (b) all predicates are p-computable on $B$; (c) all functions are p-computable functions on $B$; (d) the equality relation on $B$ is also p-computable.

**(a) P-computability of the base set $B$.**

The set $S = \bigcup_{i \in \mathbb{N}} S_i$ is the smallest fixed point of the $\Delta_0^p$-operator $\Gamma_S$. By the PAG-theorem (Theorem 2.4), $S$ is a $\Delta_0^p$-set.

An arbitrary element $b \in B$ is either an element of $S$, or is representable as a term-encoded list $\langle \underline{t}, s_1, \ldots, s_m \rangle$ where $s_1, \ldots, s_m \in S$ and $t$ is a term of signature $\sigma$. The set of all valid term-encoded elements is itself an inductively generated $\Delta_0^p$-set (cf. Corollary 2 in (7): the set of terms of any finite signature is a $\Delta_0^p$-set). Membership $\langle \underline{t}, s_1, \ldots, s_m \rangle \in B$ can be verified by:

- Checking $s_1, \ldots, s_m \in S$ (p-computable by the PAG-theorem);
- Checking that $\underline{t}$ encodes a valid term of $\sigma$ (p-computable by (7), Corollary 2);
- Verifying arity consistency.

Each step is p-computable, so $B$ is a $\Delta_0^p$-set.

**(b) P-computability of predicates.**

The predicates $P_1, \ldots, P_k$ on $\mathfrak{B}$ are extended from the finite structures $\mathfrak{M}_i$ via the $\Delta_0^p$-operator $\Gamma_{P_1,\ldots,P_k}$. The truth set of each $P_j$ on $B$ is the smallest fixed point of this operator. By the PAG-theorem (Theorem 2.4), each predicate is a $\Delta_0^p$-set.

**(c) P-computability of functions.**

For an $m_j$-ary function $f_j$ and elements $b_1, \ldots, b_{m_j} \in B$, each $b_i$ is either in $S$ or has a term representation $\underline{t_i}$. Computing $f_j(b_1, \ldots, b_{m_j})$ requires evaluating a composition of term applications. This defines a recursive function on the hereditary-finite list structure $HW(\mathfrak{M})$.

We verify that this recursive evaluation satisfies the conditions of a p-computable FGNF-system (Definition 2.5):

- *Initial function:* On base elements from $M_0$, $f_j$ is p-computable by assumption.

- *Generative term families:* For each $f_j$, the generative family $\mathcal{T}_{f_j}$ consists of terms $f_j(x_1, \ldots, x_{m_j})$ where arguments are recursively evaluated sub-terms.
- *Functional boundary:* The condition $|f_j(\bar{a})| \leq |\mathrm{list}(\bar{a})|$ directly gives the output size bound

By the FPAG-theorem (Theorem 2.6), the recursively defined evaluation functions $f_1^*, \ldots, f_n^*$ are p-computable.

**(d) P-computability of equality.**

Two elements $\underline{t_1}$ and $\underline{t_2}$ in $B$ are equal in $\mathfrak{B}$ if and only if they evaluate to the same element. Since all functions $f_j^*$ are p-computable, we can compute the values $t_1(s_1, \ldots, s_{n_1})$ and $t_2(s_1', \ldots, s_{n_2}')$ in polynomial time and compare the results. Hence, equality on $B$ is a $\Delta_0^p$-relation. $\qquad\square$

**Remark 3.7.** *For purely relational signatures $\sigma = \{P_1, \ldots, P_k\}$ (no function symbols), the theorem simplifies: conditions (2) and (5) become vacuous, all elements are generators, $B = S$, and the proof reduces to parts (a) and (b) only.*

### 3.4 Universality Theorem

We can also formulate a result establishing that under the same conditions, certain natural queries on the Fraïssé limit are polynomially decidable.

**Theorem 3.8** (P-Decidability of Bounded Formulas). *Let $\mathfrak{B}$ be the p-computable Fraïssé limit obtained from Theorem 3.6. Then for any $\Delta_0$-formula $\varphi(\bar{x})$ the truth of $\varphi(\bar{a})$ for $\bar{a} \in B$ is decidable in polynomial time.*

*Proof.* By the PAG-theorem ((7), Corollaries 3–4), the set of $\Delta_0$-formulas and conditional terms of any finite signature is a $\Delta_0^p$-set. The evaluation of a $\Delta_0$-formula on a p-computable model is itself p-computable, since all quantifiers are bounded by terms whose values can be computed in polynomial time (using the FPAG-theorem for function evaluation), and the predicates are $\Delta_0^p$-sets. $\qquad\square$

## 4 Corollaries: Classical Fraïssé Limits

We now demonstrate that four fundamental Fraïssé limits admit p-computable presentations.

### 4.1 The Random Graph (Rado Graph)

**Corollary 4.1** (P-Computability of the Random Graph). *The Rado graph $\mathcal{R}$ admits a natural p-computable presentation.*

*Proof.* **Signature.** $\sigma = \{R^{(2)}\}$ (relational, so Remark 3.7 applies).

**Step 1: Chain construction.** Set $\mathfrak{M}_0 = (\{\underline{v_0}\}, \emptyset)$ where $\underline{v_0} = \langle\square\rangle$ is the encoding of the initial vertex.

At stage $i \geq 0$, given $\mathfrak{M}_i$ with base set $M_i$, construct $\mathfrak{M}_{i+1}$ as follows. For each subset $A \subseteq M_i$, introduce a new vertex $v_A$ with:
$$R(v_A, a) \iff a \in A, \qquad R(a, v_A) \iff a \in A, \qquad \text{for all } a \in M_i.$$
Set $M_{i+1} = M_i \cup \{v_A \mid A \subseteq M_i\}$, and extend $R$ to include the new edges. This ensures the extension property: for any disjoint finite $U, V \subset M_i$, the vertex $v_U$ is adjacent to all of $U$ and none of $V$.

**Step 2: Encoding.** We encode a new vertex $v_A$ introduced at stage $i + 1$ as:
$$v_A = \langle \underbrace{\langle \cdots \langle \square \rangle \cdots \rangle}_{i+1 \text{ brackets}}, a_1, a_2, \ldots, a_m \rangle$$

where $A = \{a_1, \ldots, a_m\} \subseteq M_i$ is listed in some canonical order, and the first component $\langle^{i+1}\square\rangle^{i+1}$ serves as a *stage marker* with nesting depth $i + 1$. The nesting depth is recovered as $\mathrm{r}(\langle^{i+1}\square\rangle^{i+1}) = i + 1$. Observe that $\mathrm{r}(v_A) = i + 2$ since all $a_j$ have rank $\leq i + 1$.

The cumulative generator set is $S_i = \bigcup_{j \leq i} G(\mathfrak{M}_j)$, and the base set of the limit is $B = S = \bigcup_{i \in \mathbb{N}} S_i$.

**Step 3: Verification of the $\Delta_0^p$-operator for generators.** Define the predicate $\mathrm{Vertex}$ selecting elements of $B$. The generative family is:

$$\mathcal{F}_{\mathrm{Vertex}^+} = \left\{ \varphi_{n,k} : \; (\mathrm{r}(x_1) = k) \; \wedge \; \bigwedge_{j=2}^{n} \Big[ \mathrm{Vertex}(x_j) \; \wedge \; (\mathrm{r}(x_j) < k) \Big] \; \wedge \bigwedge_{2 \leq j < l \leq n} (x_j \neq x_l) \;\middle|\; n \geq 1, \; k \geq 1 \right\}$$

Here: $x_1$ encodes the stage marker $\langle {}^k\square\rangle^k$; the variables $x_2,\ldots,x_n$ enumerate the elements of $A$; the rank condition $\mathrm{r}(x_j) < k$ ensures all neighbors come from previous stages; the inequality conditions ensure the list $A$ has no repetitions.

We verify the $\Delta_0^p$-operator conditions:

1. *p-computable base model:* $\mathrm{HW}(\mathfrak{M}_0)$ is a p-computable structure.
2. *Positivity, quantifier-freeness, predicate separability:* Each $\varphi_{n,k}$ is quantifier-free, Vertex enters positively, and each variable appears in at most one predicate.
3. *Uniqueness:* An element $v_A = \langle\langle {}^k\square\rangle^k, a_1,\ldots,a_m\rangle$ is generated by exactly one formula $\varphi_{m+1,k}$ with $x_1 = \langle {}^k\square\rangle^k$ and $\{x_2,\ldots,x_{m+1}\} = A$.
4. *p-computability of $\gamma$:* Given $w \in \Sigma^*$, the function $\gamma(w)$ parses $w$: extract the first component, compute its rank $k$ in time $O(|w|)$, extract the remaining components and verify they are valid vertices of rank $< k$, and return the appropriate formula $\varphi_{n,k}$ (or *false* if $w$ is malformed). All steps are polynomial.

By the PAG-theorem, $S = B$ is a $\Delta_0^p$-set.

**Step 4: Verification of the $\Delta_0^p$-operator for the edge predicate.** For $a,b \in B$, the edge relation $R$ is determined by the encoding:
$$R(a,b) \iff \mathrm{Vertex}(a) \wedge \mathrm{Vertex}(b) \wedge \big(a \in b \vee b \in a\big)$$

Checking list membership is p-computable: scan the components of the longer list, comparing each to the shorter element. The time is $O(|a| \cdot |b|)$, which is polynomial.

This gives a $\Delta_0^p$-operator for $R$ with the generative family:
$$\mathcal{F}_{R^+} = \big\{\psi : \mathrm{Vertex}(x_1) \wedge \mathrm{Vertex}(x_2) \wedge (x_1 \in x_2 \vee x_2 \in x_1)\big\}$$

**Step 5: P-computability of equality.** Two elements $a,b \in B$ are equal if and only if not exists a vertex element in one of them that is not present in the other. The verification of this fact is polynomial.

All conditions of Theorem 3.6 (relational case, Remark 3.7) are satisfied. Therefore $\mathcal{R}$ is a p-computable structure. $\square$

### 4.2 COUNTABLE DENSE LINEAR ORDER WITHOUT ENDPOINTS

**Corollary 4.2** (P-Computability of the $\eta$-Order). *The countable dense linear order without endpoints $(\mathbb{Q}, \leq)$ admits a p-computable presentation.*

*Proof.* **Signature.** $\sigma = \{\leq^{(2)}\}$ (relational, so Remark 3.7 applies).

**Step 1: Chain construction.** Set $\mathfrak{M}_0 = (\{\underline{a_0}\}, \leq)$ where $\underline{a_0} = \langle\square\rangle$ and $a_0 \leq a_0$.

At stage $i \geq 0$, given $\mathfrak{M}_i$ with linearly ordered base set $M_i = \{c_1 < c_2 < \cdots < c_{N_i}\}$, construct $\mathfrak{M}_{i+1}$ by:

1. For each consecutive pair $(c_j, c_{j+1})$ with $1 \leq j \leq N_i - 1$, introduce a new element $d_{c_j, c_{j+1}}$ with $c_j < d_{c_j, c_{j+1}} < c_{j+1}$.
2. Introduce a new element $d_{\perp, c_1}$ with $d_{\perp, c_1} < c_1$ (below the minimum).
3. Introduce a new element $d_{c_{N_i}, \top}$ with $c_{N_i} < d_{c_{N_i}, \top}$ (above the maximum).

Extend $\leq$ to $M_{i+1}$ in the unique way consistent with the linear order. This construction ensures density (every gap is filled) and the absence of endpoints, yielding $(\mathbb{Q}, \leq)$ in the limit.

**Step 2: Encoding.** We encode new element $d$ on the stage $i+1$ between $a$ and $b$ is encoded as:
$$d_{a,b} = \langle\, a,\ b\,\rangle$$

where $a, b \in M_i$ are the left and right neighbors ($a$ is the element immediately below $d$, $b$ is immediately above). For the element below the minimum, encode $d_{\perp, c_1} = \langle \perp, c_1\rangle$ where $\perp$ is a distinguished symbol. For the element above the maximum, encode $d_{c_{N_i}, \top} = \langle c_{N_i}, \top\rangle$.

Every element of $B = S$ is either the initial element $\underline{a_0}$ or a triple $\langle\mathrm{marker}, \mathrm{left}, \mathrm{right}\rangle$ from some stage.

**Step 3: Verification of the $\Delta_0^p$-operator for generators.** The generative family for the predicate Elem selecting elements of $B$ is:
$$\mathcal{F}_{\mathrm{Elem}^+} = \{\varphi_k : \mathrm{Elem}(x_1) \,\&\, \mathrm{Elem}(x_2) \,\&\, Nearby(x_1, x_2) \,\&\, (x_1 \neq x_2)\}$$

$$\mathcal{F}_{\text{Nearby}^+(x_1,x_2)} = \{((r(head(x_1)) \geq r(x_2))\&(Nearby(head(x_1),x_2)));$$
$$((r(x_1) < r(first(x_2)))\&(Nearby(x_1,first(x_2))));(x_1 = x_2)\}$$

**Step 4: Verification of the $\Delta_0^p$-operator for the order predicate.** Given $a, b \in B$, we need to decide $a \leq b$ in polynomial time. The key observation is that the encoding allows a recursive comparison:

$$\mathcal{F}_{\leq^+(x_1,x_2)} = \{(((r(x_1) > r(x_2))\& \leq (head(x_1),x_2));(((r(x_1) < r(x_2))\& \leq (x_1,first(x_2)));$$

$$(r(x_1) = r(x_2))\&(x_1 \neq x_2)\& \leq (head(x_1),first(x_2)); x_1 = x_2 \}$$

**Step 5: P-computability of equality.** Elements are in canonical form by construction, so equality is string comparison in linear time.

Therefore $(\mathbb{Q}, \leq)$ is p-computable. $\qquad\square$

## 4.3 COUNTABLE ATOMLESS BOOLEAN ALGEBRA

**Corollary 4.3** (P-Computability of the Atomless Boolean Algebra)**.** *The countable atomless Boolean algebra $\mathfrak{A}$ admits a p-computable presentation.*

*Proof.* **Signature.** $\sigma = \{\wedge, \vee, \neg, 0, 1\}$ (functional, so the full Theorem 3.6 applies).

**Step 1: Chain construction via iterated splitting.** We construct $\mathfrak{A}$ as the direct limit of a chain of finite Boolean algebras $\mathfrak{M}_0 \hookrightarrow \mathfrak{M}_1 \hookrightarrow \cdots$, where at each stage every atom is split into two.

*Stage 0.* $\mathfrak{M}_0 = \{0, 1\}$, the two-element Boolean algebra. Encode: $\underline{0} = \bot, \underline{1} = \top$.

*Stage $i + 1$.* Let $At(\mathfrak{M}_i) = \{a_1, \ldots, a_m\}$ be the set of atoms of $\mathfrak{M}_i$. For each atom $a_j$, introduce two new atoms $a_j^L$ and $a_j^R$ with $a_j^L \vee a_j^R = a_j$ and $a_j^L \wedge a_j^R = 0$. The new algebra $\mathfrak{M}_{i+1}$ has atoms $\{a_j^L, a_j^R \mid 1 \leq j \leq m\}$ and all elements are joins of subsets of these atoms. Every element of $\mathfrak{M}_i$ is faithfully embedded: the atom $a_j$ maps to $a_j^L \vee a_j^R$.

This gives an atomless Boolean algebra in the limit, since every nonzero element eventually splits.

**Step 2: Encoding via binary paths.** Each atom at stage $i$ corresponds to a binary string of length $i$: a path in the full binary tree. Concretely, the atom obtained by splitting $a_j^L$ (going left) and then $a_j^{RL}$ (going right then left) corresponds to the string $LRL\cdots$. We encode atoms as:

$$\text{atom:}w_1 w_2 \cdots w_i = \langle < \ldots, << \top, w_1 >, w_2 > \ldots, w_i >\rangle, \qquad w_j \in \{L, R\}.$$

An arbitrary element of $\mathfrak{M}_i$ is a join of atoms, so we encode it as the *sorted list of its atoms*:

$$b = \langle \alpha_1, \alpha_2, \ldots, \alpha_s \rangle$$

where $\alpha_1 < \alpha_2 < \cdots < \alpha_s$ in lexicographic order and each $\alpha_j$ is an atom encoding.

This encoding is *canonical*: each element has a unique representation as a sorted list of atoms.

**Step 3: Generators and the $\Delta_0^p$-operator $\Gamma_S$.** The generators at stage $i + 1$ are the new atoms $\{a_j^L, a_j^R\}$ and all new joins involving at least one new atom. The generative family for the predicate BA ("is a Boolean algebra element") is:

$$\mathcal{F}_{\text{BA}^+} = \left\{ \varphi_n : \bigwedge_{j=1}^{n} \text{Atom}(x_j) \wedge \bigwedge_{1 \leq j < l \leq n} (x_j <_{\text{lex}} x_l)\&(x_{l+1} = \square^s, \&(x_{l+1} = \square^{s*} \,\middle|\, n \geq 1 \right\}$$

where $s$ is a length of the full binary tree of the level $n = max\{r(x_i)|i \in I\}$ minus length of all atoms of the element, Atom is itself defined by a generating family:

$$\mathcal{F}_{\text{Atom}^+} = \{\psi_k : (x_{n+1} = L \vee x_{n+1} = R) \& Atom(<< \top, x_1 >, \ldots, x_n >)\} \qquad (1)$$

The conditions are verified:

1. All formulas are positive, quantifier-free, predicate-separable.
2. Uniqueness: the canonical sorted-list representation ensures each element is generated by exactly one formula.
3. $\gamma(w)$: parse $w$ to extract the tag and atom list, verify each component is a valid binary string, check sorted order. Time: $O(|w|^2)$ (pairwise comparison of atoms).

By the PAG-theorem, the predicate BA are $\Delta_0^p$-sets.

**Step 4: P-computable closure, operations and the functional boundary.**

Closure will be made term-by-element. Any combination of terms will be a closure, and there will also be a canonical encoding, which will be the domain of the functions values.

We can define on step $i$ the closure $Cl_i$: $\underline{t(\overline{a})}$ the some set of terms which we can constract.

This set is p-computable. The set of canonical elements also p-computable.

**Step 5: Functional boundary verification.**

For all functions, functional boundedness holds, which means all conditions of Theorem 3.6 are satisfied (inner embedding, functional boundary, $\Delta_0^p$-operators for generators, predicates, and canonicity). Therefore $\mathfrak{A}$ is p-computable. $\qquad\square$

### 4.4 Countable Ershov Algebra

**Corollary 4.4** (P-Computability of the Ershov Algebra). *The countable atomless Ershov algebra (distributive lattice with relative complements) admits a p-computable presentation.*

*Proof.* The construction mirrors that of the atomless Boolean algebra (Corollary 4.3) with appropriate modifications for relative complements instead of absolute complements. Specifically, the signature is $\sigma = \{\wedge, \vee, 0, 1, \backslash\}$ where $a \backslash b$ denotes the relative complement.

The binary tree encoding is adapted so that the relative complement operation $a \backslash b$ is computed by recursive traversal of the tree structure. The functional boundary condition $|a \backslash b| \leq |\langle a, b\rangle|$ is maintained by the encoding, and the FPAG-theorem applies by the same argument as in Corollary 4.3.

By Theorem 3.6, the Ershov algebra is p-computable. $\qquad\square$

## 5 Applications to Artificial Intelligence

### 5.1 Polynomial Complexity as a Reliability Criterion

In trustworthy AI systems—particularly decision-support systems for critical infrastructure, autonomous vehicles, and medical diagnostics—**polynomial computational complexity is a fundamental reliability criterion**, not merely a performance optimization. A system that may require exponential time on certain inputs is, from a safety perspective, equivalent to a system that may not terminate at all. The P-Fraïssé Theorem provides mathematical guarantees that certain classes of inductively constructed data structures can always be queried in polynomial time.

### 5.2 Digital Twins and Smart City Infrastructure

A *digital twin* of a smart city (17) is a hierarchical computational model reflecting the structure of physical urban systems:

$$\text{City} \supset \text{Districts} \supset \text{Quarters} \supset \text{Buildings} \supset \text{Apartments} \supset \text{Residents}$$

This hierarchy is naturally modeled as a Fraïssé-like limit construction: at each level, new elements (buildings, residents) are added to the existing structure, and the predicates and functions (population counts, resource usage, connectivity) are extended inductively. The P-Fraïssé Theorem guarantees that:

1. **Membership queries** ("Is this entity part of the city model?") are decidable in polynomial time.
2. **Predicate evaluation** ("Is building $X$ connected to network $Y$?") is polynomial.
3. **Aggregate functions** ("What is the total population of district $D$?") are polynomial, by the FPAG-theorem component of our proof.

As shown in (8), the recursive function `numResidents`$(w)$ that traverses the hierarchical city structure satisfies the conditions of a p-computable FGNF-system, and its polynomial complexity follows directly from the FPAG-theorem.

### 5.3 KNOWLEDGE GRAPHS AS FRAÏSSÉ LIMITS

A *knowledge graph* in an AI system can be viewed as an evolving relational structure: entities are added over time, and relations between them are established inductively. When the knowledge graph possesses the amalgamation property (different sources of knowledge can be consistently merged), it can be modeled as a Fraïssé limit. The random graph corollary (Corollary 4.1) shows that even the most "generic" such structure—one with the extension property—admits polynomial-time algorithms for basic queries.

### 5.4 FORMAL VERIFICATION AND SMART CONTRACTS

In blockchain systems, smart contracts must execute within bounded resources ("gas" in Ethereum). The FPAG-theorem, as a component of the P-Fraïssé Theorem, provides sufficient conditions for recursive functions to remain polynomial. This is directly applicable to smart contracts that process hierarchically structured data (e.g., nested financial instruments, recursive organizational structures) and must be guaranteed to terminate within polynomial bounds (18).

### 5.5 HYBRID AI ARCHITECTURE

Modern AI architectures increasingly combine large language models (LLMs) with formal reasoning engines. The P-Fraïssé Theorem contributes to this paradigm by providing a class of structures for which the formal reasoning component can guarantee polynomial-time performance. LLMs can translate natural-language queries into formal queries over the p-computable Fraïssé limit, which are then evaluated by the formal engine with guaranteed polynomial complexity—a necessary component of trustworthy AI systems (16).

## 6 MATHEMATICAL SIGNIFICANCE AND OPEN PROBLEMS

### 6.1 CONTRIBUTION TO MODEL THEORY

The P-Fraïssé Theorem establishes a polynomial-time realization result for a classical model-theoretic construction. While the existence of computable Fraïssé limits has been studied (15; 13), explicit *polynomial* bounds on their computability are new. This opens a systematic program:

> *For which Fraïssé classes does the limit admit a p-computable presentation?*

Our four corollaries provide initial data points. The natural next targets include the generic partial order, the generic tournament, the generic directed graph, and the rational Urysohn space.

### 6.2 CONTRIBUTION TO COMPLEXITY THEORY

The interaction between the PAG-theorem (for predicates) and the FPAG-theorem (for functions) provides a general methodology for establishing polynomial bounds on inductively defined structures. This methodology extends beyond Fraïssé limits to any scenario where structures are built through iterated extensions satisfying appropriate effectiveness and boundedness conditions.

### 6.3 OPEN PROBLEMS

1. **Relaxation of the functional boundary.** Can the condition $|f_j(\bar{a})| \leq |\text{list}(\bar{a})|$ be relaxed to $|f_j(\bar{a})| \leq C \cdot |\text{list}(\bar{a})|^p$ while still maintaining overall p-computability of the limit?
2. **Non-chain constructions.** Can the P-Fraïssé Theorem be extended to Fraïssé limits constructed via more general directed systems (not necessarily chains)?
3. **Quantitative bounds.** What is the optimal polynomial degree for membership and predicate evaluation in specific Fraïssé limits? Is there a uniform bound?
4. **Automorphism groups.** The automorphism group of a Fraïssé limit is a key invariant. Do the polynomial bounds on the structure induce polynomial bounds on certain group-theoretic operations?
5. **Topological dynamics.** Fraïssé theory connects to topological dynamics via extreme amenability and Ramsey theory. Do the polynomial bounds on the structures have implications for the computational complexity of these connections?

## 7 CONCLUSION

We have established the P-Fraïssé Theorem, providing sufficient conditions for Fraïssé limits to be p-computable. The proof synthesizes the PAG-theorem (for inductive predicate extensions), the FPAG-theorem (for recur-

sive function definitions), and the classical theory of Fraïssé limits. Four concrete corollaries demonstrate p-computability for the random graph, the dense linear order, the atomless Boolean algebra, and the Ershov algebra.

These results are significant across mathematics, computer science, and artificial intelligence. For mathematicians, they initiate a systematic complexity-theoretic study of classical model-theoretic constructions. For computer scientists, they provide provable polynomial-time guarantees for a natural class of inductively defined structures. For AI practitioners, they ensure that hierarchical models of real-world systems—from smart cities to knowledge graphs—can be reliably queried within polynomial time, a foundational requirement for trustworthy AI.

**Acknowledgments.** The authors thank the research team at the Artificial Intelligence Center of Novosibirsk State University and the Sobolev Institute of Mathematics for valuable discussions.

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
