# OpenReview forum: "Polynomial-Time Fraisse Limits via Fixed Point Theorems: From Model Theory to Trustworthy AI"
_mathai.club/MathAI/2026/Conference — 2026 Oral_

### Official Review · Reviewer_L13z · 2026-03-12
**The review of "Polynomial-Time Fraisse Limits via Fixed Point Theorems: From Model Theory to Trustworthy AI"**

**Rating:** 7
**Confidence:** 4

**Review:**

This paper studies polynomial-time computability of Fraïssé limits constructed from chains of finite structures. By combining techniques from computable model theory and complexity theory, the authors establish sufficient conditions under which Fraïssé limits remain polynomial-time computable. The central result, the P-Fraïssé theorem, relies on the polynomial analogue of Gandy’s fixed point theorem (PAG) and its functional extension (FPAG). The paper further demonstrates that several classical Fraïssé limits, including the random graph, the countable dense linear order without endpoints, the atomless Boolean algebra, and the Ershov algebra, admit p-computable presentations under natural encodings.

Strengths:
- The paper develops a rigorous theoretical framework connecting Fraïssé theory with polynomial-time computability.
- The use of the PAG and FPAG theorems provides a systematic method for analyzing complexity bounds of inductively constructed structures.
- The P-Fraïssé theorem gives clear sufficient conditions for polynomial-time computability of Fraïssé limits.
- The paper demonstrates the applicability of the framework to several classical structures from model theory.
- The work highlights potential applications in trustworthy AI and knowledge representation systems where guaranteed polynomial-time reasoning is important.

Suggestions for improvement:

The paper could be further strengthened by:
- providing additional discussion of the complexity bounds in concrete constructions;
- clarifying some technical steps in the example constructions;
- further elaborating the connection between the theoretical results and practical AI systems.

Final Recommendation:

POSTED / Accept with minor revision

Overall, the paper presents a mathematically strong contribution connecting model theory, complexity theory, and formal reasoning in AI-related systems. The results provide a useful perspective on polynomial-time computability of inductively constructed structures and may stimulate further research at the intersection of logic and artificial intelligence.

---

### Official Review · Reviewer_UBEp · 2026-03-13
**This article is devoted to establishing the polynomial computability of the Fresset limits of fundamental constructs in model theory. The authors formulate and prove a central result in the form of a theorem stating that, under certain conditions, the Fresset limit itself is a computable structure.**

**Rating:** 6
**Confidence:** 3

**Review:**

Several passages in the article raise questions.
1) There are symbols not explained in the text, for example, in Corollary 4.3.
2) The detail of the proofs in the corollaries raises questions. The proof of Theorem 3.8 consists of a single sentence. This seems insufficient.
3) In some places, the definitions contain unclosed parentheses (in 4.3), there are many typos, and there are unfinished sentences.


The article's main problem is that the connection between its mathematical results and its claimed applications to AI is purely declarative. The keywords "trustworthy AI," "digital twins," and "smart cities" in the title and abstract set expectations that the text fails to fulfill.

---

### Official Review · Reviewer_EXKh · 2026-03-13
**Polynomial-Time Fraisse Limits via Fixed Point Theorems: From Model Theory to Trustworthy AI.    Good paper, accept.**

**Rating:** 7
**Confidence:** 4

**Review:**

The article is devoted to Fraisse's theorem and related mathematical tools for solving a number of important problems in modeling theory and providing a canonical method for constructing countable homogeneous structures as limits of directed systems of finite structures. The received results bridge classical model-theoretic constructions with the computational complexity requirements of trustworthy artificial intelligence, digital twin infrastructures for smart cities, and formally verified decision-support systems, where polynomial computational complexity serves as a fundamental reliability criterion. The reviewer believes that the article was written by a highly qualified specialist whose report will be of interest to many conference participants.
1.	Mathematical Rigor: high.
2.	Novelty & Contribution: good.
3.	Relevance to MathAI: very high.
4.	Technical Quality: good.
5.	Clarity & Presentation: good.
6.	AI-Generation Risk: very low.

---

### Decision · Program_Chairs · 2026-03-14

**Decision:**

Accept (Oral)

**Comment:**

Dear Author(s),

On behalf of the Program Committee of the International Conference on Mathematics of Artificial Intelligence (MathAI 2026), we are pleased to inform you that your paper has been accepted for an oral presentation at MathAI 2026.

Your paper was evaluated through a rigorous two-stage review process involving both automated screening and expert review by members of the Program Committee. The reviewers recognized the quality and contribution of your work.

Presentation details:

- Format: Oral presentation (15–20 minutes + 5 minutes Q&A)
- Mode: You may present either in person (offline) at the conference venue in Sirius, Russia, or remotely via Zoom. Please indicate your preferred mode when confirming your participation.
- Conference dates: Marh 30 - April 3, 2026
- Website: https://mathai.club

Next steps:

1. Please confirm your participation and presentation mode by replying to this email mathai.club@yandex.ru no later than March 15, 2026 18:00 Moscow time.
2. If you plan to attend in person, the organizing committee will provide accommodation details separately.
3. Please prepare your final camera-ready manuscript according to the formatting guidelines available at https://mathai.club and upload it to OpenReview by March 15, 2026 18:00 Moscow time.

Should you have any questions regarding the program, logistics, or your presentation slot, please do not hesitate to contact us.

We look forward to your contribution to MathAI 2026.

With kind regards,

MathAI 2026 Program Committee
International Conference on Mathematics of Artificial Intelligence
https://mathai.club
OpenReview: https://openreview.net/group?id=mathai.club/MathAI/2026/Conference
Telegram: https://t.me/MathAI_club
Email: mathai.club@yandex.ru